# Noisy Exemplars Make Large Language Models More Robust: A Domain-Agnostic Behavioral Analysis

**Hongyi Zheng**
New York University
hz2212@nyu.edu

**Abulhair Saparov**
New York University
as17582@nyu.edu

## Abstract

Recent advances in prompt engineering enable large language models (LLMs) to solve multi-hop logical reasoning problems with impressive accuracy. However, there is little existing work investigating the robustness of LLMs with few-shot prompting techniques. Therefore, we introduce a systematic approach to test the robustness of LLMs in multi-hop reasoning tasks via domain-agnostic perturbations. We include perturbations at multiple levels of abstractions (e.g. lexical perturbations such as typos, and semantic perturbations such as the inclusion of intermediate reasoning steps in the questions) to conduct behavioral analysis on the LLMs. Throughout our experiments, we find that models are more sensitive to certain perturbations such as replacing words with their synonyms. We also demonstrate that increasing the proportion of perturbed exemplars in the prompts improves the robustness of few-shot prompting methods.

## 1 Introduction

Large language models (LLMs) achieve human-like performance on many natural language processing tasks after few-shot learning due to increasing scale (Kaplan et al., 2020). However, they often struggle in conducting multi-hop reasoning tasks after standard prompting (Rae et al., 2021). Recently, multiple prompt engineering methods such as chain-of-thought prompting (Wei et al., 2023), zero-shot prompting (Kojima et al., 2023) and least-to-most-prompting (Zhou et al., 2023) have led to significant empirical improvements in these tasks.

Despite these signs of progress, there is an important drawback in recent studies: the datasets used in these experiments are often idealized, noise-free, and rather distinct from examples that LLMs encounter in real applications, which put the generalizability of these prompting methods into question since applications of LLMs in practice are often noisy, containing errors, redundant or irrelevant sentences, utilizing out-of-distribution vocabulary, etc. For instance, Cheng et al. (2018) show that

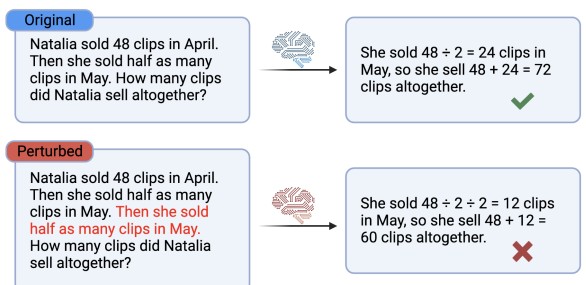

Figure 1: A simple repetition may result in LLM producing incorrect solution.

very small amount of common lexical perturbations such as word replacement and deletion could result in drastic change in machine translation results.

There are few existing studies investigating the robustness of these prompting schemes through behavioral experiments on perturbed examples. Thus in our research, we create a selection of domain-agnostic tests to investigate the robustness of state-of-the-art prompting methods. Our two main goals are: (1) to compare and contrast the performance of prompting methods with respect to various perturbations, and (2) to explore empirical approaches that may improve their robustness.[1]

## 2 Related Work

### 2.1 Prompt Engineering

Contemporary prompt engineering methods that aim to improve LLMs' reasoning performance stem from chain-of-thought prompting (CoT) proposed by Wei et al. (2023), which draws inspiration from the earlier work of Ling et al. (2017) with the key idea of augmenting standard few-shot prompting with a chain-of-thought (i.e. a description of the reasoning steps that lead to the answer). CoT improves LLM's performance in a wide range of reasoning tasks. Nevertheless, recent research shows that it may lead to inconsistent reasoning steps

---

[1]Our code is open source and available at https://github.com/Hiroki39/Noisy-Exemplars-Make-Large-Language-Models-More-Robust

| **Original** | **Typo** | **Synonym** | **Repetition** | **Shortcut** |
|---|---|---|---|---|
| Elsa has 5 cars. Anna has 2 more cars than Elsa. How many cars do they have together? | Elsa has 5 cars. Anna has 2 more cras than Elsa. How many cars do thye have together? | Elsa has 5 cars. Anna has 2 more vehicles than Elsa. How many automobiles do they have together? | Elsa has 5 cars. Anna has 2 more cars than Elsa. Elsa has 5 cars. How many cars do they have together? | Elsa has 5 cars. Anna has 2 more cars than Elsa. Anna has 7 cars. How many cars do they have together? |

Figure 2: Examples for each type of perturbation test. A more detailed example is shown in Table 1.

and thus worse performance under certain circumstances (Ye and Durrett, 2022a), which highlights the necessity to conduct further behavioral analyses to identify such circumstances and find ways to mitigate this issue.

CoT inspired a few subsequent prompting techniques. Zero-shot prompting (0CoT) proposed by Kojima et al. (2023) requires significantly less human engineering to generate prompts compared with the original approach. Least-to-most prompting (LtM) proposed by Zhou et al. (2023) decomposes complex reasoning tasks into easier subproblems to improve the model performance. Selection-Inference prompting (Creswell et al., 2023), LAMBADA (Kazemi et al., 2023), and tree-of-thought prompting (Yao et al., 2023) further break down the problem so that the LLM is queried for each step of the reasoning. Lastly, self-consistency prompting (Wang et al., 2023) uses sampling and aggregation techniques to diversify reasoning paths and increase the chance of deriving correct answers.

These methods have been shown to be effective in increasing model accuracy under noise-free environments. Our work, on the other hand, focuses on investigating the robustness of these methods when a variety of perturbations are present.

## 2.2 Behavioral Testing

The concept of behavioral testing (also known as black-box testing) is first proposed by Beizer and Wiley (1996) as an effective approach to probing large software or computer systems. Ribeiro et al. (2020) brought many of these insights to the testing of NLP models and proposed CHECKLIST, a comprehensive domain-agnostic methodology that embraces the benefits of challenge sets such as systematic control of test examples (Belinkov and Glass, 2019) while avoiding their drawbacks such as the fact that challenge sets are artificial and do not resemble real data.

Recent behavioral analysis of LLMs has revealed their deficiencies when handling domain-specific perturbations. LLMs are shown to be sensitive to domain-specific perturbations in reading comprehension (Jia and Liang, 2017), text classification (Gan and Mori, 2023), as well as logical

reasoning tasks (Ye and Durrett, 2022b). Some tests investigate LLM consistency via replacing real concept nouns with fictional ones (Saparov and He, 2023) and adding semantically related yet logically irrelevant distractor sentences (Saparov et al., 2023). While those studies mainly focus on domain-specific perturbations, or on synthetic settings, our work evaluates the prompt engineering methods' robustness against *domain-agnostic* perturbations applied to more realistic data, across more levels of abstraction, which are more widespread in the practical use of LLMs.

## 3 Method

We aim to conduct domain-agnostic analysis with tests that span multiple levels of abstraction, are easy to automate, while still closely resembling examples in real applications. Thus, we use the following four perturbation tests:

**Typo**. We test whether the model's output is sensitive to typing errors in its input by introducing typos. To be robust to this perturbation, LLMs cannot rely on copying problem sentences with repeated words to produce reasoning chains. Typos are introduced by randomly swapping one character with its adjacent character within a token with probability $0.1$ given that the token has more than one character and is not numeric.

**Synonym**. We test whether models could recognize semantically similar tokens that refer to the same object of interest by replacing some words with synonyms. To be robust to this perturbation, LLMs should not exploit lexical spurious correlations. Operationally, every noun and verb token is replaced by one of its synonyms within its WordNet (Miller, 1992) synsets with probability $0.2$.[2]

**Repetition**. We also test whether models are robust to relevant but redundant information by duplicating a sentence in the input. To be robust to this perturbation, the model must ignore the redundant sentence, or utilize it to complete the CoT. Operationally, we randomly choose a sentence within the problem text other than the question sentence (i.e.

---

[2]higher than **Typo** probability to ensure the number of perturbed tokens is comparable with **Typo** test

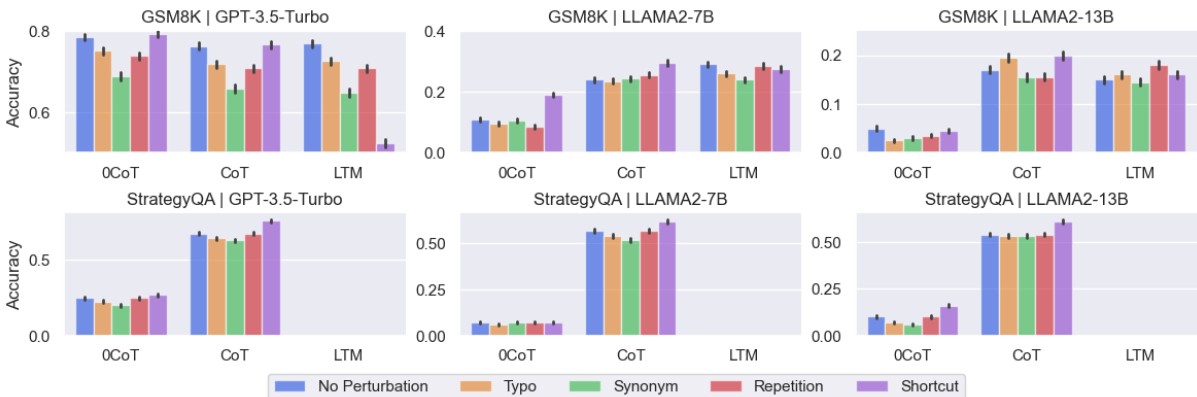

Figure 3: Perturbations on the test question vs accuracy under various combinations of datasets and prompting approaches. 95% confidence intervals are shown.

the last sentence), and insert a copy of it before the last sentence. This minimizes the risk of breaking coreference links (Jia and Liang, 2017).

**Shortcut.** We test whether model behavior is affected if an intermediate result is given in the question description. The LLM may take advantage of this extra information to expedite reasoning. Conversely, the LLM could also be confused, as it would serve as redundant information if the LLM first ignored this directly given intermediate result but later derived it. Operationally, we extract the first reasoning step for the problem (first hop for CoT and 0CoT; first sub-problem and its corresponding answer for LTM) and insert it before the question sentence.

## 4 Experimental Setup

### 4.1 Model and Dataset

The GSM8K (Cobbe et al., 2021) and StrategyQA (Geva et al., 2021) datasets are used for all experiments. The relatively simple problem statements in these datasets facilitate the creation of adversarial examples for testing. The two datasets provide the intermediate reasoning steps required by the few-shot prompting methods. For the choice of LLMs, GPT-3.5-Turbo (Brown et al., 2020), LLaMA2 7B and LLaMA2 13B models are used in our experiments. All models are open source with weights available for public use.

### 4.2 Prompting Methods

We experiment with CoT, 0CoT, and LTM. Our prompt design closely follows Zhou et al. (2023), Kojima et al. (2023), and Shi et al. (2023). For few-shot prompting techniques (i.e. CoT and LTM), each input prompt includes either one or eight exemplar problems as well as their corresponding

solution, followed by the test problem. The exemplars are created from the problems within GSM8K either with or without perturbations to allow us better understand how perturbing the exemplars would affect the performance of the LLMs in answering the test problem. For 0CoT, the test question is directly presented without exemplars but instead followed by "A: *Let's think step by step*". To facilitate answer extraction, a high-level instruction, "*End your response with 'The answer is <answer>'*" is prepended to every prompt. We do not apply LTM prompting for StrategyQA as it does not provide high-quality decompositions for each question, which are required by LTM prompting.

## 5 Results

We carry out two experiments. In the first experiment, we evaluate the effect of perturbations of the test question on LLM performance under different prompting methods. In the second experiment, we perturb exemplar questions and vary the proportion of perturbed exemplars for few-shot prompting techniques to investigate whether an increasing share of perturbed exemplars would lead to better robustness against the perturbations in the test question.

### 5.1 Do Perturbations in the Test Question Affect Prompting Performance?

In this experiment, we investigate the effect of perturbations on prompting methods with original unperturbed exemplars. For CoT and LTM trials, one unperturbed question and its associated reasoning process are provided as the exemplar, followed by the perturbed test question, whereas no exemplar is provided for 0CoT trials. The results are illustrated in Figure 3. We draw a few conclusions from the results:

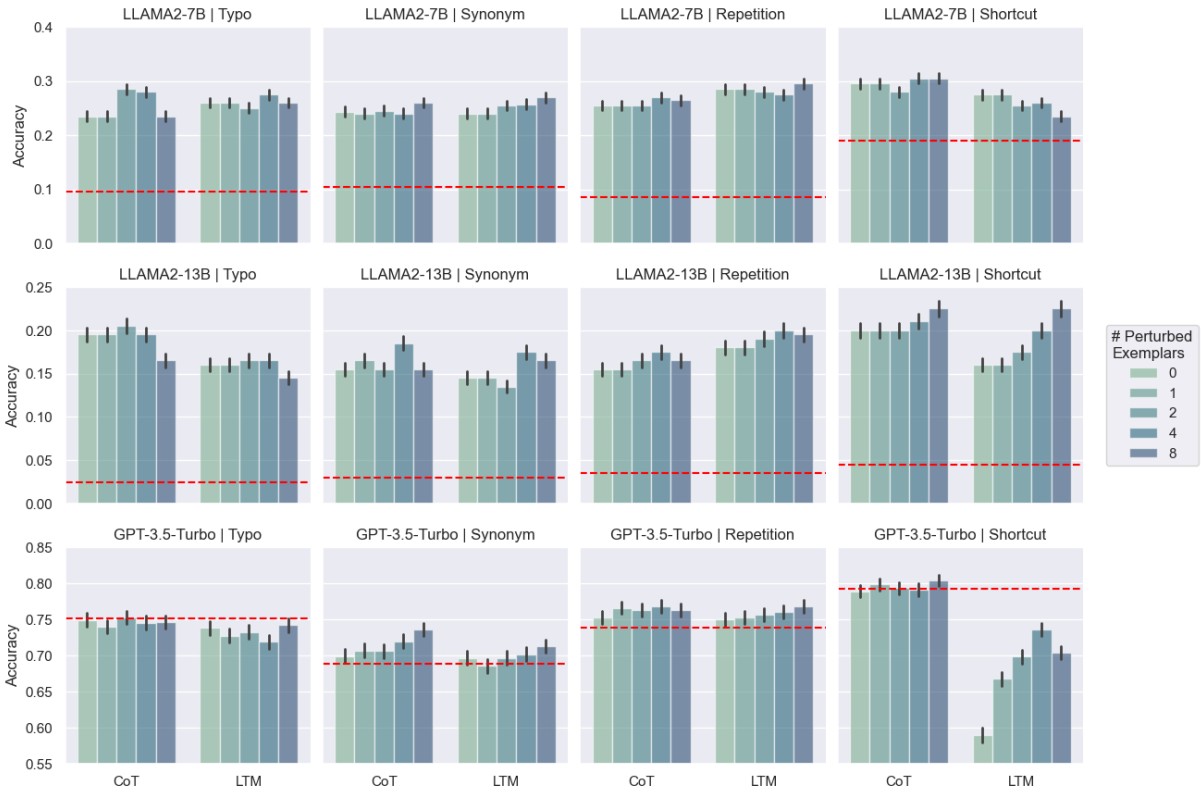

Figure 4: Number of perturbed in-context exemplars vs accuracy for GSM8K trials. Note that the total number of in-context exemplars is always 8. Dashed red lines indicate the accuracy of 0CoT in subsection 5.1. 95% confidence intervals are shown.

**In general, prompting methods are most susceptible to Synonym replacement**. When conducting the **Synonym** tests, the accuracy of the LLM falls below the baseline performance across trials. For instance, for GPT-3.5-Turbo on the GSM8K dataset, **Synonym** perturbation lowers the accuracy by $0.097 \sim 0.122$ (green bars in the top-left subplot of Figure 3) as compared to the baseline (blue bars), which are larger decreases as compared to the **Typo** ($0.034 \sim 0.044$, orange bars) and **Repetition** tests ($0.046 \sim 0.062$, red bars). This might result from low-frequency phrases and sentences created by the **Synonym** perturbation, as every token is substituted with its synonym regardless of the adjacent tokens. Many substitutions would likely create grammatically correct yet infrequent sentences and phrases (e.g. "She eats three *repasts* per day") which might lead to worse performance.

## 5.2 Do Perturbations in the Exemplars Affect Few-shot Prompting Performance?

Following the previous experiment, a question of interest naturally arises: would perturbing the exemplars during the few-shot learning process improves the robustness of few-shot prompting methods? To answer this question, we conduct another

experiment for two few-shot prompting methods CoT and LTM. In each trial, 8 exemplars are presented to the LLM, with a proportion of them (0, 1, 2, 4, or 8 out of 8) being perturbed. Then, the perturbed test question is presented to the LLM. The results are shown in Figure 4 and Figure 5. We find the following key observations:

**Increasing the proportion of perturbed exemplars improves few-shot prompting performance, except for the Typo perturbation.** From the results, the accuracy for both few-shot prompting methods trends upwards when the number of perturbed exemplars presented to the LLM increases aside from LLaMA2-13B on the shortcut perturbation, demonstrating the evidence that the LLM is able to adapt to the perturbations through the few-shot learning process. Take GPT-3.5-Turbo on GSM8K dataset as an example, if excluding trials with the **Typo** perturbation, increasing the number of perturbed exemplars from 0 to 8 (lightest bars versus darkest bars in the third row of Figure 4) results in an average of $0.035$ increase in accuracy, with the minimum increase of $0.011$ and the maximum of $0.114$. The performance of the LLM in many trials surpasses the benchmark set by the 0CoT method in subsection 5.1 with the max-

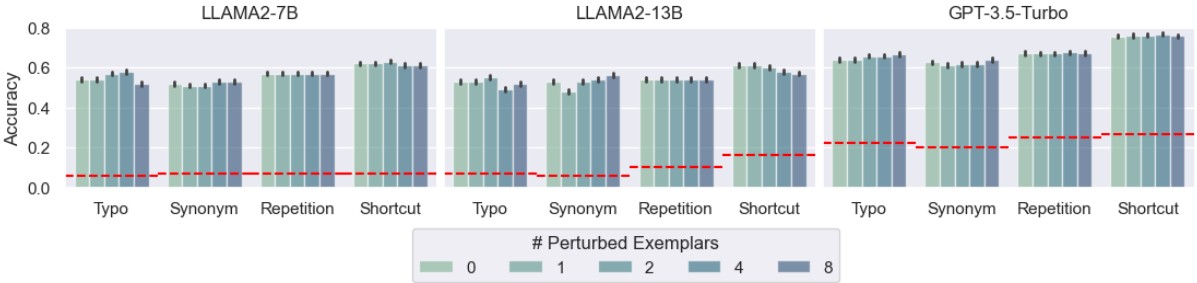

Figure 5: Number of perturbed in-context exemplars vs accuracy for StrategyQA trials. Note that we do not apply LTM on StrategyQA and all bars represent CoT results. Dashed red lines indicate the accuracy of 0CoT in subsection 5.1. 95% confidence intervals are shown.

imum advantage of $0.048$ achieved by CoT after few-shot learning with all eight exemplars with the **Synonym** perturbation. **Typo** trials turn out to be the exception, where increasing the number of perturbed exemplars does not lead to an improvement in accuracy. We speculate that this results from the fact that typos are much more common in the pretraining dataset as compared to other perturbations, which offsets the benefits of few-shot learning on perturbed exemplars.

This experiment suggests that perturbing in-context exemplars may serve as a more efficient alternative to augmenting pretraining with perturbations. It is much cheaper and easier to introduce perturbed exemplars at inference time rather than in pretraining.

## 6  Discussion and Future Work

In conclusion, through two experiments, we conducted an investigation of the robustness of state-of-the-art prompting schemes via a series of domain-agnostic perturbation tests. Our first experiment revealed the robustness of 0CoT prompting versus few-shot prompting methods when the few-shot exemplars are unperturbed. Our second experiment demonstrated that perturbing few-shot exemplars led to notable improvements in robustness to perturbations in the test question, which is valuable in real applications. Suppose a user knows in advance that the questions of interest will be subject to some certain kind of perturbation (e.g. mathematics problems scraped from the online forums which are subject to typos and uneliminated HTML tags, or questions asked by beginner-level English speakers that may contain grammatical errors). The user can then prepend exemplars perturbed in a similar fashion to improve the robustness of the LLM on their test examples. This can easily be applied in real-world use cases as all perturbations discussed above are easily automated.

There are a number of directions for future work available: the first is to explore the effect of more comprehensive perturbations, such as presenting the LLM with a problem statement containing multiple possible lines of reasoning that each lead to the correct conclusion, or adding some semantically related but logically irrelevant extra information to the problem statement (Shi et al., 2023). Another direction is to explore the influence of several other variables on the robustness of the prompting methods, such as the model size, the number of hops required to answer the problem, whether the problem is counterfactual, etc. Lastly, we observe that in some cases, though the LLM is able to produce the correct answer under perturbation, the model is more likely to produce sentences with errors as shown in Table 1 (e.g. "Janet lay 16 eggs per day"). Further studying the relationship between perturbations in the model's input and the errors in the output would be illuminating.

## Limitations

We acknowledge several limitations of our work. Firstly, due to the constraints in computational power, available time, and budget, we only experiment with a relatively small set of LLMs and datasets, and our results could benefit from experiments on an increased number of models and datasets with more diverse tasks. Secondly, some perturbations might introduce unintended side effects. For instance, in the **Repetition** test, adding repeated information might not always be idempotent, as repeating sentences like *A man walked into the bar* might alter the correct answer to the problem. These nuances and subtleties might require more detailed manual inspection.

## Reproducibility Statement

In our experiments, a fixed random seed (42) is applied over all trials to ensure the reproducibility of perturbed questions and exemplars, and the

sampling temperature is set to $0$ so that the model behavior is as deterministic as possible. The experiment output files as well as the analysis codes are available in our GitHub repository (`www.github.com/Hiroki39/Noisy-Exemplars-Make-Large-Language-Models-More-Robust`).

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

# Appendices

## A    Test Question and Model Output Example

| Input Question | Model Output | | |
|---|---|---|---|
| | **CoT** | **0CoT** | **LtM** |
| **Original** Janet's ducks lay 16 eggs per day. She eats three for breakfast every morning and bakes muffins for her friends every day with four. She sells the remainder at the farmers' market daily for $2 per fresh duck egg. How much in dollars does she make every day at the farmers' market? | Janet uses 3+4=7 eggs every day for breakfast and muffins. So she has 16-7=9 eggs left to sell at the farmers' market. She makes 9*2=$18 every day at the farmers' market. The answer is 18. | Janet has 16 eggs per day. She eats 3 for breakfast and uses 4 for muffins, which leaves her with 9 eggs. She sells these 9 eggs at $2 per egg, which means she makes $18 per day at the farmers' market. The answer is $18. | How many eggs does Janet use for breakfast and muffins every day? Janet uses 3+4=7 eggs every day. How many eggs does she have left to sell? So she has 16-7=9 eggs left to sell every day. How much money does she make selling the eggs? Selling each egg for $2, she makes 9*2=$18 every day at the farmers' market. The answer is 18. |
| **Typo** Janet's ducks aly 16 eggs per day. She eats three for breakfast every omrning and bakes muffnis for her friends every day with four. She sells the remainder at the farmers' market daily ofr $2 per fresh duck egg. How much in dollars does she make every day at the farmers' market? | Janet uses 3 eggs for breakfast and 4 for muffins, so she uses 3+4=7 eggs per day. That means she has 16-7=9 eggs left to sell at the farmers' market. Selling each egg for $2, she makes 9*2=$18 every day at the farmers' market. The answer is 18. | Jante has 16 eggs per day, she eats 3 for breakfast and uses 4 for muffins, which leaves her with 9 eggs. She sells these 9 eggs at $2 per egg, which means she makes $18 every day at the farmers' market. The answer is $18. | How many eggs does Janet use for breakfast every day? Janet uses 3 eggs for breakfast every day. How many eggs does Janet use for muffins every day? Janet uses 4 eggs for muffins every day. How many eggs does Janet have left to sell? Janet has 16 - 3 - 4 = 9 eggs left to sell. How much money does Janet make every day at the farmers' market? Janet makes 9 x $2 = $18 every day at the farmers' market. The answer is $18. |
| **Synonym** Janet's ducks lay 16 eggs per day. She eats three for breakfast every morning and bakes muffins for her friends every day with four. She sells the residue at the farmers' market daily for $2 per fresh duck egg. How much in dollars does she make every sidereal day at the farmers' market? | Janet lays 16 eggs per day, and she eats 3 + 4 = 7 eggs per day. So, she has 16 - 7 = 9 eggs left to sell at the farmers' market. She makes 9 x $2 = $18 every day at the farmers' market. The answer is 18. | Janet has 16 duck eggs per day. She eats 3 for breakfast and bakes muffins with 4, which leaves her with 9 eggs. She sells the remaining 9 eggs at the farmers' market for $2 each, which means she makes $18 per day. The answer is $18. | How many orchids does Janet have left after breakfast and baking muffins? Janet has 16 - 3 - 4 = 9 orchids left. How much money does she make from selling these orchids? She makes 9 x $2 = $18 from selling the orchids. How much money does Janet make every day at the farmers' market? So Janet makes $18 every day at the farmers' market. The answer is 18. |
| **Repetition** Janet's ducks lay 16 eggs per day. She eats three for breakfast every morning and bakes muffins for her friends every day with four. She sells the remainder at the farmers' market daily for $2 per fresh duck egg. Janet's ducks lay 16 eggs per day. How much in dollars does she make every day at the farmers' market? | Janet uses 3+4=7 eggs every day for breakfast and muffins. So she has 16-7=9 eggs left to sell at the farmers' market. She makes 9*2=$18 every day at the farmers' market. The answer is 18. | Janet has 16 eggs per day. She eats 3 for breakfast and uses 4 for muffins, which leaves her with 9 eggs. She sells these 9 eggs at $2 per egg, which means she makes $18 every day at the farmers' market. The answer is $18. | How many eggs does Janet use for breakfast and muffins every day? Janet uses 3+4=7 eggs every day. How many eggs does she have left to sell? So she has 16-7=9 eggs left to sell every day. How much money does she make selling the eggs? Selling each egg for $2, she makes 9*2=$18 every day at the farmers' market. The answer is 18. |

| **Shortcut** | | | |
|---|---|---|---|
| Janet's ducks lay 16 eggs per day. She eats three for breakfast every morning and bakes muffins for her friends every day with four. She sells the remainder at the farmers' market daily for $2 per fresh duck egg. Janet sells 16 - 3 - 4 = 9 duck eggs a day. How much in dollars does she make every day at the farmers' market? | Janet sells 9 duck eggs a day, so she makes 9 x 2 =18 every day at the farmers' market. The answer is $18. | Janet sells 9 duck eggs a day, so she makes 9 x $2 = $18 every day at the farmers' market. The answer is $18. | How many eggs does Janet sell? Janet sells 9 duck eggs a day. How much in dollars does she make every day at the farmers' market? She makes 9 x 2 =18 every day at the farmers' market. The answer is 18. |

Table 1: An example of different perturbations and corresponding model outputs. The first column illustrates the original and perturbed example test question and the next three columns show the model outputs with CoT, 0CoT, and LTM respectively.