# OpenReview forum: "Noisy Exemplars Make Large Language Models More Robust: A Domain-Agnostic Behavioral Analysis"
_EMNLP/2023/Conference — EMNLP 2023 Main_

### Official Review · Reviewer_dKHZ · 2023-08-02

**Soundness:** 3

**Excitement:**

3: Ambivalent: It has merits (e.g., it reports state-of-the-art results, the idea is nice), but there are key weaknesses (e.g., it describes incremental work), and it can significantly benefit from another round of revision. However, I won't object to accepting it if my co-reviewers champion it.

**Paper Topic And Main Contributions:**

The paper introduces an approach to assess the robustness of large language models (LLMs) in multi-hop arithmetic reasoning tasks using domain-agnostic perturbations. The authors apply perturbations at various levels of abstraction including lexical and semantic changes (typos, repetitions, shortcuts and synonyms) to conduct an in-depth analysis of the LLM behavior. The authors mention that although recent advances in prompt engineering have shown impressive accuracy in multi-hop logical reasoning, their paper is one of the first to explore the robustness of LLMs with few-shot prompting techniques. The experiments in the paper demonstrate that increasing the proportion of perturbed examples in the prompts enhances the robustness of few-shot prompting methods thus helping attain greater resilience in LLMs' performance on multi-hop reasoning tasks.

**Reasons To Accept:**

- The authors present a novel research direction that might be of great value in the LLM research space as their adaptation in zero-shot settings increases.

**Reasons To Reject:**

- The scope of the paper is quite limited, no conclusions can be made using the data the authors provide since they test preliminary perturbations only on a single language model.
- The authors make some conclusions based on the results they see on a single dataset using a single model, first, the performance variations don't look very significant to me, second, although the variation is significant I would argue that the evidence is too modest to make any conclusions.

**Reproducibility:**

5: Could easily reproduce the results.

**Reviewer Confidence:**

4: Quite sure. I tried to check the important points carefully. It's unlikely, though conceivable, that I missed something that should affect my ratings.

---

> ### Author Rebuttal · Authors · 2023-08-29
>
> We thank the reviewer for their helpful and constructive feedback. We appreciate any additional comments or concerns.
>
> The main concern of the reviewer is the limited generalizability of our results, since our experiments were limited to a single language model (gpt-3.5-turbo) and a single dataset (GSM8K). We address this concern by performing additional experiments on LLaMA-2 7B and 13B, as well as on StrategyQA [1], a commonsense question-answering dataset. We chose StrategyQA as it is distinct from GSM8K, and tests different reasoning abilities of LLMs. The experiment on the open model LLaMA-2 also helps to facilitate reproducibility.
>
> The results for the experiment where 8 unperturbed few-shot examples are given for each perturbed test example (Fig 3 in the paper) are shown below:
>
> | GSM8K 0CoT   | ChatGPT  |   LLaMA-2 13B  |   LLaMA-2 7B |
> |:-------------------|:---------------:|:--------------------:|:-------------------:|
> | no perturbation |     0.78       |           0.11        |         0.08        |
> | typo                  |     0.75       |          0.10         |         0.04        |
> | synonym         |      0.69      |           0.11        |          0.04       |
> | repetition         |      0.74      |           0.09        |         0.05        |
> | shortcut          |     0.79       |           0.19       |          0.02        |
>
> | GSM8K   CoT   | ChatGPT  |   LLaMA-2 13B  |   LLaMA-2 7B |
> |:-------------------|:---------------:|:--------------------:|:-------------------:|
> | no perturbation |   0.78         |          0.24         |       0.16          |
> | typo                 |    0.75        |           0.24        |        0.19         |
> | synonym         |    0.70        |           0.24        |        0.15         |
> | repetition         |    0.75        |           0.26        |        0.14         |
> | shortcut           |    0.79       |            0.30        |        0.18         |
>
> | GSM8K   LTM  |  ChatGPT  |   LLaMA-2 13B |    LLaMA-2 7B |
> |:-------------------|:---------------:|:--------------------:|:-------------------:|
> |no perturbation  |    0.77        |        0.29        |         0.15          |
> |typo                   |     0.74       |        0.26        |         0.19          |
> |synonym           |     0.70       |        0.24        |         0.14          |
> |repetition          |     0.75        |        0.29        |         0.22          |
> |shortcut            |     0.59        |        0.28        |         0.17          |
>
> | StratQA 0CoT  | ChatGPT   |  LLaMA-2 13B   |  LLaMA-2 7B  |
> |:-------------------|:---------------:|:--------------------:|:-------------------:|
> | no perturbation |    0.25       |        0.07            |    0.10          |
> | typo                  |    0.22       |         0.06           |     0.07          |
> | synonym          |     0.20      |         0.07           |     0.06          |
> | repetition          |    0.25       |         0.07           |     0.10          |
> | shortcut           |     0.27      |          0.07         |      0.16          |
>
> | StratQA  CoT  |  ChatGPT  |   LLaMA-2 13B  |   LLaMA-2 7B
> |:-------------------|:---------------:|:--------------------:|:-------------------:|
> | no perturbation |   0.67       |            0.57     |           0.54      |
> | typo                |      0.64      |              0.54   |             0.53   |
> | synonym        |     0.63        |            0.52      |          0.53     |
> | repetition        |     0.67        |            0.57      |          0.54    |
> | shortcut         |     0.76         |           0.62       |          0.61     |
>
> Note that in the camera-ready version, all results mentioned here will be rendered as bar plots with confidence intervals, similar to Figs 3 and 4, making use of the extra page allowance.
>
> The results for the experiment where the proportion of perturbed few-shot examples is varied (Fig 4 in the paper) is shown below:
>
> GSM8K Typo CoT:
> | num perturbed exemplars |    0   |    1   |   2   |    4   |    8    |
> |:----------------------------------|:------:|:------:|:-----:|:------:|:------:|
> | ChatGPT                           | 0.75 | 0.74 | 0.75 | 0.75 | 0.75 |
> | LLaMA-2 13B                    | 0.24 | 0.24 | 0.29 | 0.28 | 0.24 |
> | LLaMA-2 7B                      | 0.20 | 0.20 | 0.21 | 0.20 | 0.17 |
>
> GSM8K Synonym CoT:
> | num perturbed exemplars |    0   |    1   |   2   |    4   |    8    |
> |:----------------------------------|:------:|:------:|:-----:|:------:|:------:|
> | ChatGPT                           | 0.70 | 0.71 | 0.71 | 0.72 | 0.74 |
> | LLaMA-2 13B                    | 0.24 | 0.24 | 0.25 | 0.24 | 0.26 |
> | LLaMA-2 7B                      | 0.15 | 0.17 | 0.16 | 0.19 | 0.16 |
>
> GSM8K Repetition CoT:
> | num perturbed exemplars |    0   |    1   |   2   |    4   |    8    |
> |:----------------------------------|:------:|:------:|:-----:|:------:|:------:|
> | ChatGPT                           | 0.75 | 0.77 | 0.76 | 0.77 | 0.76 |
> | LLaMA-2 13B                    | 0.26 | 0.26 | 0.26 | 0.27 | 0.26 |
> | LLaMA-2 7B                      | 0.16 | 0.16 | 0.17 | 0.18 | 0.17 |
>
> GSM8K Shortcut CoT:
> | num perturbed exemplars |    0   |    1   |   2   |    4   |    8    |
> |:----------------------------------|:------:|:------:|:-----:|:------:|:------:|
> | ChatGPT                           | 0.79 | 0.80 | 0.79 | 0.79 | 0.80 |
> | LLaMA-2 13B                    | 0.30 | 0.30 | 0.28 | 0.31 | 0.31 |
> | LLaMA-2 7B                      | 0.20 | 0.20 | 0.20 | 0.21 | 0.22 |
>
> GSM8K Typo LTM:
> | num perturbed exemplars |    0   |    1   |   2   |    4   |    8    |
> |:----------------------------------|:------:|:------:|:-----:|:------:|:------:|
> | ChatGPT                           | 0.74 | 0.73 | 0.73 | 0.72 | 0.74 |
> | LLaMA-2 13B                    | 0.26 | 0.26 | 0.25 | 0.28 | 0.26 |
> | LLaMA-2 7B                      | 0.16 | 0.16 | 0.17 | 0.17 | 0.15 |
>
> GSM8K Synonym LTM:
> | num perturbed exemplars |    0   |    1   |   2   |    4   |    8    |
> |:----------------------------------|:------:|:------:|:-----:|:------:|:------:|
> | ChatGPT                           | 0.70 | 0.69 | 0.70 | 0.70 | 0.71 |
> | LLaMA-2 13B                    | 0.24 | 0.24 | 0.26 | 0.26 | 0.27 |
> | LLaMA-2 7B                      | 0.15 | 0.15 | 0.14 | 0.18 | 0.17 |
>
> GSM8K Repetition LTM:
> | num perturbed exemplars |    0   |    1   |   2   |    4   |    8    |
> |:----------------------------------|:------:|:------:|:-----:|:------:|:------:|
> | ChatGPT                           | 0.75 | 0.75 | 0.76 | 0.76 | 0.77 |
> | LLaMA-2 13B                    | 0.29 | 0.29 | 0.28 | 0.28 | 0.30 |
> | LLaMA-2 7B                      | 0.18 | 0.18 | 0.19 | 0.20 | 0.20 |
>
> GSM8K Shortcut LTM:
> | num perturbed exemplars |    0   |    1   |   2   |    4   |    8    |
> |:----------------------------------|:------:|:------:|:-----:|:------:|:------:|
> | ChatGPT                           | 0.59 | 0.67 | 0.70 | 0.74 | 0.70 |
> | LLaMA-2 13B                    | 0.28 | 0.28 | 0.26 | 0.26 | 0.24 |
> | LLaMA-2 7B                      | 0.16 | 0.16 | 0.18 | 0.20 | 0.23 |
>
> Strategy QA Typo CoT:
> | num perturbed exemplars |    0   |    1   |   2   |    4   |    8    |
> |:----------------------------------|:------:|:------:|:-----:|:------:|:------:|
> | ChatGPT                           | 0.64 | 0.64 | 0.66 | 0.66 | 0.67 |
> | LLaMA-2 13B                    | 0.54 | 0.54 | 0.57 | 0.58 | 0.52 |
> | LLaMA-2 7B                      | 0.53 | 0.53 | 0.55 | 0.49 | 0.52 |
>
> Strategy QA Synonym CoT:
> | num perturbed exemplars |    0   |    1   |   2   |    4   |    8    |
> |:----------------------------------|:------:|:------:|:-----:|:------:|:------:|
> | ChatGPT                           | 0.63 | 0.61 | 0.62 | 0.62 | 0.64 |
> | LLaMA-2 13B                    | 0.52 | 0.51 | 0.51 | 0.53 | 0.53 |
> | LLaMA-2 7B                      | 0.53 | 0.48 | 0.53 | 0.54 | 0.56 |
>
> Strategy QA Repetition CoT:
> | num perturbed exemplars |    0   |    1   |   2   |    4   |    8    |
> |:----------------------------------|:------:|:------:|:-----:|:------:|:------:|
> | ChatGPT                           | 0.67 | 0.67 | 0.67 | 0.68 | 0.67 |
> | LLaMA-2 13B                    | 0.57 | 0.57 | 0.57 | 0.57 | 0.57 |
> | LLaMA-2 7B                      | 0.54 | 0.54 | 0.54 | 0.54 | 0.54 |
>
> Strategy QA Shortcut CoT:
> | num perturbed exemplars |    0   |    1   |   2   |    4   |    8    |
> |:----------------------------------|:------:|:------:|:-----:|:------:|:------:|
> | ChatGPT                           | 0.76 | 0.76 | 0.76 | 0.77 | 0.76 |
> | LLaMA-2 13B                    | 0.62 | 0.61 | 0.63 | 0.61 | 0.61 |
> | LLaMA-2 7B                      | 0.61 | 0.61 | 0.60 | 0.58 | 0.57 |
>
> Consistent with the previous results on ChatGPT, LLaMA 7B generally seems to favor having more perturbed examples than fewer. Also consistent with earlier results, CoT is more robust against perturbations than LTM prompting. We propose to replace Figs 3 and 4 with the data from the new experiments shown above (rendered as bar plots) and add additional text to Sec 5 to better contextualize the results, mentioning that the models are more sensitive to perturbations on GSM8K than StrategyQA. We will also qualify the claims in the Introduction and Discussion sections accordingly.
>
> [1] Mor Geva, Daniel Khashabi, Elad Segal, Tushar Khot, Dan Roth, & Jonathan Berant (2021). Did Aristotle Use a Laptop? A Question Answering Benchmark with Implicit Reasoning Strategies. TACL.

---

### Official Review · Reviewer_Scw6 · 2023-08-05

**Soundness:** 3

**Excitement:**

3: Ambivalent: It has merits (e.g., it reports state-of-the-art results, the idea is nice), but there are key weaknesses (e.g., it describes incremental work), and it can significantly benefit from another round of revision. However, I won't object to accepting it if my co-reviewers champion it.

**Paper Topic And Main Contributions:**

This paper focuses on the robustness of large language models (LLMs) in multi-hop arithmetic reasoning tasks using few-shot prompting techniques. The main contributions of the paper include introducing a systematic approach to test the robustness of LLMs through domain-agnostic perturbations, conducting behavioral analysis on LLMs using perturbations at multiple levels of abstractions, and comparing the performance of different prompting methods with respect to various perturbations. The paper also explores empirical approaches to improve the robustness of LLMs in these tasks.

**Reasons To Accept:**

1. Systematic Approach: This paper introduces a systematic approach to test the robustness of large language models (LLMs) in multi-hop arithmetic reasoning tasks. It includes domain-agnostic perturbations at multiple levels of abstractions, such as lexical and semantic perturbations, to conduct behavioral analysis on LLMs.

2. Comparison of Prompting Methods: The paper compares and contrasts the performance of different prompting methods, including chain-of-thought prompting (COT), zero-shot prompting, and least-to-most prompting (LTM), with respect to various perturbations. This provides valuable insights into the strengths and weaknesses of these methods.

3. Empirical Approaches to Improve Robustness: The paper explores empirical approaches to improve the robustness of prompting methods. It demonstrates that increasing the proportion of perturbed exemplars in the prompts improves the robustness of few-shot prompting methods.

**Reasons To Reject:**

1. Limited Generalizability: The paper acknowledges that the experiments were only conducted with GPT-3 and GSM8K dataset, which limits the generalizability of the conclusions. This raises concerns about the applicability of the findings to other language models and datasets.

2. Unintended Side Effects: The paper mentions that some perturbations, such as adding repeated information, might introduce unintended side effects. This could potentially alter the correct answer to the problem and affect the reliability of the model's output.

3. Need for Manual Inspection: The nuances and subtleties of the perturbations used in the experiments might require more detailed manual inspection. This suggests that the automated evaluation alone may not capture all the potential issues and limitations of the proposed methods.

**Reproducibility:**

3: Could reproduce the results with some difficulty. The settings of parameters are underspecified or subjectively determined; the training/evaluation data are not widely available.

**Reviewer Confidence:**

3: Pretty sure, but there's a chance I missed something. Although I have a good feel for this area in general, I did not carefully check the paper's details, e.g., the math, experimental design, or novelty.

---

> ### Author Rebuttal · Authors · 2023-08-29
>
> We thank the reviewer for their helpful and constructive feedback. We appreciate any additional comments or concerns.
>
> The main concern of the reviewer is the limited generalizability of our results, since our experiments were limited to a single language model (gpt-3.5-turbo) and a single dataset (GSM8K). We address this concern by performing additional experiments on LLaMA-2 7B and 13B, as well as on StrategyQA [1], a commonsense question-answering dataset. We chose StrategyQA as it is distinct from GSM8K, and tests different reasoning abilities of LLMs. The experiment on the open model LLaMA-2 also helps to facilitate reproducibility.
>
> The results for the experiment where 8 unperturbed few-shot examples are given for each perturbed test example (Fig 3 in the paper) are shown below:
>
> | GSM8K 0CoT   | ChatGPT  |   LLaMA-2 13B  |   LLaMA-2 7B |
> |:-------------------|:---------------:|:--------------------:|:-------------------:|
> | no perturbation |     0.78       |           0.11        |         0.08        |
> | typo                  |     0.75       |          0.10         |         0.04        |
> | synonym         |      0.69      |           0.11        |          0.04       |
> | repetition         |      0.74      |           0.09        |         0.05        |
> | shortcut          |     0.79       |           0.19       |          0.02        |
>
> | GSM8K   CoT   | ChatGPT  |   LLaMA-2 13B  |   LLaMA-2 7B |
> |:-------------------|:---------------:|:--------------------:|:-------------------:|
> | no perturbation |   0.78         |          0.24         |       0.16          |
> | typo                 |    0.75        |           0.24        |        0.19         |
> | synonym         |    0.70        |           0.24        |        0.15         |
> | repetition         |    0.75        |           0.26        |        0.14         |
> | shortcut           |    0.79       |            0.30        |        0.18         |
>
> | GSM8K   LTM  |  ChatGPT  |   LLaMA-2 13B |    LLaMA-2 7B |
> |:-------------------|:---------------:|:--------------------:|:-------------------:|
> |no perturbation  |    0.77        |        0.29        |         0.15          |
> |typo                   |     0.74       |        0.26        |         0.19          |
> |synonym           |     0.70       |        0.24        |         0.14          |
> |repetition          |     0.75        |        0.29        |         0.22          |
> |shortcut            |     0.59        |        0.28        |         0.17          |
>
> | StratQA 0CoT  | ChatGPT   |  LLaMA-2 13B   |  LLaMA-2 7B  |
> |:-------------------|:---------------:|:--------------------:|:-------------------:|
> | no perturbation |    0.25       |        0.07            |    0.10          |
> | typo                  |    0.22       |         0.06           |     0.07          |
> | synonym          |     0.20      |         0.07           |     0.06          |
> | repetition          |    0.25       |         0.07           |     0.10          |
> | shortcut           |     0.27      |          0.07         |      0.16          |
>
> | StratQA  CoT  |  ChatGPT  |   LLaMA-2 13B  |   LLaMA-2 7B
> |:-------------------|:---------------:|:--------------------:|:-------------------:|
> | no perturbation |   0.67       |            0.57     |           0.54      |
> | typo                |      0.64      |              0.54   |             0.53   |
> | synonym        |     0.63        |            0.52      |          0.53     |
> | repetition        |     0.67        |            0.57      |          0.54    |
> | shortcut         |     0.76         |           0.62       |          0.61     |
>
> Note that in the camera-ready version, all results mentioned here will be rendered as bar plots with confidence intervals, similar to Figs 3 and 4, making use of the extra page allowance.
>
> The results for the experiment where the proportion of perturbed few-shot examples is varied (Fig 4 in the paper) is shown below:
>
> GSM8K Typo CoT:
> | num perturbed exemplars |    0   |    1   |   2   |    4   |    8    |
> |:----------------------------------|:------:|:------:|:-----:|:------:|:------:|
> | ChatGPT                           | 0.75 | 0.74 | 0.75 | 0.75 | 0.75 |
> | LLaMA-2 13B                    | 0.24 | 0.24 | 0.29 | 0.28 | 0.24 |
> | LLaMA-2 7B                      | 0.20 | 0.20 | 0.21 | 0.20 | 0.17 |
>
> GSM8K Synonym CoT:
> | num perturbed exemplars |    0   |    1   |   2   |    4   |    8    |
> |:----------------------------------|:------:|:------:|:-----:|:------:|:------:|
> | ChatGPT                           | 0.70 | 0.71 | 0.71 | 0.72 | 0.74 |
> | LLaMA-2 13B                    | 0.24 | 0.24 | 0.25 | 0.24 | 0.26 |
> | LLaMA-2 7B                      | 0.15 | 0.17 | 0.16 | 0.19 | 0.16 |
>
> GSM8K Repetition CoT:
> | num perturbed exemplars |    0   |    1   |   2   |    4   |    8    |
> |:----------------------------------|:------:|:------:|:-----:|:------:|:------:|
> | ChatGPT                           | 0.75 | 0.77 | 0.76 | 0.77 | 0.76 |
> | LLaMA-2 13B                    | 0.26 | 0.26 | 0.26 | 0.27 | 0.26 |
> | LLaMA-2 7B                      | 0.16 | 0.16 | 0.17 | 0.18 | 0.17 |
>
> GSM8K Shortcut CoT:
> | num perturbed exemplars |    0   |    1   |   2   |    4   |    8    |
> |:----------------------------------|:------:|:------:|:-----:|:------:|:------:|
> | ChatGPT                           | 0.79 | 0.80 | 0.79 | 0.79 | 0.80 |
> | LLaMA-2 13B                    | 0.30 | 0.30 | 0.28 | 0.31 | 0.31 |
> | LLaMA-2 7B                      | 0.20 | 0.20 | 0.20 | 0.21 | 0.22 |
>
> GSM8K Typo LTM:
> | num perturbed exemplars |    0   |    1   |   2   |    4   |    8    |
> |:----------------------------------|:------:|:------:|:-----:|:------:|:------:|
> | ChatGPT                           | 0.74 | 0.73 | 0.73 | 0.72 | 0.74 |
> | LLaMA-2 13B                    | 0.26 | 0.26 | 0.25 | 0.28 | 0.26 |
> | LLaMA-2 7B                      | 0.16 | 0.16 | 0.17 | 0.17 | 0.15 |
>
> GSM8K Synonym LTM:
> | num perturbed exemplars |    0   |    1   |   2   |    4   |    8    |
> |:----------------------------------|:------:|:------:|:-----:|:------:|:------:|
> | ChatGPT                           | 0.70 | 0.69 | 0.70 | 0.70 | 0.71 |
> | LLaMA-2 13B                    | 0.24 | 0.24 | 0.26 | 0.26 | 0.27 |
> | LLaMA-2 7B                      | 0.15 | 0.15 | 0.14 | 0.18 | 0.17 |
>
> GSM8K Repetition LTM:
> | num perturbed exemplars |    0   |    1   |   2   |    4   |    8    |
> |:----------------------------------|:------:|:------:|:-----:|:------:|:------:|
> | ChatGPT                           | 0.75 | 0.75 | 0.76 | 0.76 | 0.77 |
> | LLaMA-2 13B                    | 0.29 | 0.29 | 0.28 | 0.28 | 0.30 |
> | LLaMA-2 7B                      | 0.18 | 0.18 | 0.19 | 0.20 | 0.20 |
>
> GSM8K Shortcut LTM:
> | num perturbed exemplars |    0   |    1   |   2   |    4   |    8    |
> |:----------------------------------|:------:|:------:|:-----:|:------:|:------:|
> | ChatGPT                           | 0.59 | 0.67 | 0.70 | 0.74 | 0.70 |
> | LLaMA-2 13B                    | 0.28 | 0.28 | 0.26 | 0.26 | 0.24 |
> | LLaMA-2 7B                      | 0.16 | 0.16 | 0.18 | 0.20 | 0.23 |
>
> Strategy QA Typo CoT:
> | num perturbed exemplars |    0   |    1   |   2   |    4   |    8    |
> |:----------------------------------|:------:|:------:|:-----:|:------:|:------:|
> | ChatGPT                           | 0.64 | 0.64 | 0.66 | 0.66 | 0.67 |
> | LLaMA-2 13B                    | 0.54 | 0.54 | 0.57 | 0.58 | 0.52 |
> | LLaMA-2 7B                      | 0.53 | 0.53 | 0.55 | 0.49 | 0.52 |
>
> Strategy QA Synonym CoT:
> | num perturbed exemplars |    0   |    1   |   2   |    4   |    8    |
> |:----------------------------------|:------:|:------:|:-----:|:------:|:------:|
> | ChatGPT                           | 0.63 | 0.61 | 0.62 | 0.62 | 0.64 |
> | LLaMA-2 13B                    | 0.52 | 0.51 | 0.51 | 0.53 | 0.53 |
> | LLaMA-2 7B                      | 0.53 | 0.48 | 0.53 | 0.54 | 0.56 |
>
> Strategy QA Repetition CoT:
> | num perturbed exemplars |    0   |    1   |   2   |    4   |    8    |
> |:----------------------------------|:------:|:------:|:-----:|:------:|:------:|
> | ChatGPT                           | 0.67 | 0.67 | 0.67 | 0.68 | 0.67 |
> | LLaMA-2 13B                    | 0.57 | 0.57 | 0.57 | 0.57 | 0.57 |
> | LLaMA-2 7B                      | 0.54 | 0.54 | 0.54 | 0.54 | 0.54 |
>
> Strategy QA Shortcut CoT:
> | num perturbed exemplars |    0   |    1   |   2   |    4   |    8    |
> |:----------------------------------|:------:|:------:|:-----:|:------:|:------:|
> | ChatGPT                           | 0.76 | 0.76 | 0.76 | 0.77 | 0.76 |
> | LLaMA-2 13B                    | 0.62 | 0.61 | 0.63 | 0.61 | 0.61 |
> | LLaMA-2 7B                      | 0.61 | 0.61 | 0.60 | 0.58 | 0.57 |
>
> Consistent with the previous results on ChatGPT, LLaMA 7B generally seems to favor having more perturbed examples than fewer. Also consistent with earlier results, CoT is more robust against perturbations than LTM prompting. We propose to replace Figs 3 and 4 with the data from the new experiments shown above (rendered as bar plots) and add additional text to Sec 5 to better contextualize the results, mentioning that the models are more sensitive to perturbations on GSM8K than StrategyQA. We will also qualify the claims in the Introduction and Discussion sections accordingly.
>
> [1] Mor Geva, Daniel Khashabi, Elad Segal, Tushar Khot, Dan Roth, & Jonathan Berant (2021). Did Aristotle Use a Laptop? A Question Answering Benchmark with Implicit Reasoning Strategies. TACL.

---

### Official Review · Reviewer_5nja · 2023-08-11

**Soundness:** 3

**Excitement:**

3: Ambivalent: It has merits (e.g., it reports state-of-the-art results, the idea is nice), but there are key weaknesses (e.g., it describes incremental work), and it can significantly benefit from another round of revision. However, I won't object to accepting it if my co-reviewers champion it.

**Paper Topic And Main Contributions:**

This paper investigates the robustness of large language models (LLMs) in multi-hop arithmetic reasoning tasks using domain-agnostic perturbations. The authors introduce a systematic approach (Typo, Synonym, Repetition Shortcut) to test the robustness of LLMs and compare the performance of different prompting methods under various perturbations.

**Questions For The Authors:**

1. The author claim that one of the goal is `to explore empirical approaches that may improve their robustness.`, but I did not see much discussion on this. Could you provide more information on it?

**Reasons To Accept:**

1. The paper is well-motivated and clearly written.
2. Although the experiment is not comprehensive, this paper investigate the effect of several perturbation techniques on arithmetic reasoning tasks.

**Reasons To Reject:**

1. Experiments are limited: Due to the computational power, only using gpt-3.5-turbo is acceptable, but at least experiment on two datasets which would make the conclusion more reliable. Also, if more domain can be tested (commonsense reasoning, symbolic reasoning) would be better.
2. Setting is limited: To analyse the robustness of LLMs, it is expected to see more (e.g. written style perturbations, order perturbations)

**Reproducibility:**

5: Could easily reproduce the results.

**Reviewer Confidence:**

4: Quite sure. I tried to check the important points carefully. It's unlikely, though conceivable, that I missed something that should affect my ratings.

---

> ### Author Rebuttal · Authors · 2023-08-29
>
> ## Rebuttal
>
> We thank the reviewer for their helpful and constructive feedback. We appreciate any additional comments or concerns.
>
> The main concern of the reviewer is the limited generalizability of our results, since our experiments were limited to a single language model (gpt-3.5-turbo) and a single dataset (GSM8K). We address this concern by performing additional experiments on LLaMA-2 7B and 13B, as well as on StrategyQA [1], a commonsense question-answering dataset. We chose StrategyQA as it is distinct from GSM8K, and tests different reasoning abilities of LLMs. The experiment on the open model LLaMA-2 also helps to facilitate reproducibility.
>
> The results for the experiment where 8 unperturbed few-shot examples are given for each perturbed test example (Fig 3 in the paper) are shown below:
>
> | GSM8K 0CoT   | ChatGPT  |   LLaMA-2 13B  |   LLaMA-2 7B |
> |:-------------------|:---------------:|:--------------------:|:-------------------:|
> | no perturbation |     0.78       |           0.11        |         0.08        |
> | typo                  |     0.75       |          0.10         |         0.04        |
> | synonym         |      0.69      |           0.11        |          0.04       |
> | repetition         |      0.74      |           0.09        |         0.05        |
> | shortcut          |     0.79       |           0.19       |          0.02        |
>
> | GSM8K   CoT   | ChatGPT  |   LLaMA-2 13B  |   LLaMA-2 7B |
> |:-------------------|:---------------:|:--------------------:|:-------------------:|
> | no perturbation |   0.78         |          0.24         |       0.16          |
> | typo                 |    0.75        |           0.24        |        0.19         |
> | synonym         |    0.70        |           0.24        |        0.15         |
> | repetition         |    0.75        |           0.26        |        0.14         |
> | shortcut           |    0.79       |            0.30        |        0.18         |
>
> | GSM8K   LTM  |  ChatGPT  |   LLaMA-2 13B |    LLaMA-2 7B |
> |:-------------------|:---------------:|:--------------------:|:-------------------:|
> |no perturbation  |    0.77        |        0.29        |         0.15          |
> |typo                   |     0.74       |        0.26        |         0.19          |
> |synonym           |     0.70       |        0.24        |         0.14          |
> |repetition          |     0.75        |        0.29        |         0.22          |
> |shortcut            |     0.59        |        0.28        |         0.17          |
>
> | StratQA 0CoT  | ChatGPT   |  LLaMA-2 13B   |  LLaMA-2 7B  |
> |:-------------------|:---------------:|:--------------------:|:-------------------:|
> | no perturbation |    0.25       |        0.07            |    0.10          |
> | typo                  |    0.22       |         0.06           |     0.07          |
> | synonym          |     0.20      |         0.07           |     0.06          |
> | repetition          |    0.25       |         0.07           |     0.10          |
> | shortcut           |     0.27      |          0.07         |      0.16          |
>
> | StratQA  CoT  |  ChatGPT  |   LLaMA-2 13B  |   LLaMA-2 7B
> |:-------------------|:---------------:|:--------------------:|:-------------------:|
> | no perturbation |   0.67       |            0.57     |           0.54      |
> | typo                |      0.64      |              0.54   |             0.53   |
> | synonym        |     0.63        |            0.52      |          0.53     |
> | repetition        |     0.67        |            0.57      |          0.54    |
> | shortcut         |     0.76         |           0.62       |          0.61     |
>
> Note that in the camera-ready version, all results mentioned here will be rendered as bar plots with confidence intervals, similar to Figs 3 and 4, making use of the extra page allowance.
>
> The results for the experiment where the proportion of perturbed few-shot examples is varied (Fig 4 in the paper) is shown below:
>
> GSM8K Typo CoT:
> | num perturbed exemplars |    0   |    1   |   2   |    4   |    8    |
> |:----------------------------------|:------:|:------:|:-----:|:------:|:------:|
> | ChatGPT                           | 0.75 | 0.74 | 0.75 | 0.75 | 0.75 |
> | LLaMA-2 13B                    | 0.24 | 0.24 | 0.29 | 0.28 | 0.24 |
> | LLaMA-2 7B                      | 0.20 | 0.20 | 0.21 | 0.20 | 0.17 |
>
> GSM8K Synonym CoT:
> | num perturbed exemplars |    0   |    1   |   2   |    4   |    8    |
> |:----------------------------------|:------:|:------:|:-----:|:------:|:------:|
> | ChatGPT                           | 0.70 | 0.71 | 0.71 | 0.72 | 0.74 |
> | LLaMA-2 13B                    | 0.24 | 0.24 | 0.25 | 0.24 | 0.26 |
> | LLaMA-2 7B                      | 0.15 | 0.17 | 0.16 | 0.19 | 0.16 |
>
> GSM8K Repetition CoT:
> | num perturbed exemplars |    0   |    1   |   2   |    4   |    8    |
> |:----------------------------------|:------:|:------:|:-----:|:------:|:------:|
> | ChatGPT                           | 0.75 | 0.77 | 0.76 | 0.77 | 0.76 |
> | LLaMA-2 13B                    | 0.26 | 0.26 | 0.26 | 0.27 | 0.26 |
> | LLaMA-2 7B                      | 0.16 | 0.16 | 0.17 | 0.18 | 0.17 |
>
> GSM8K Shortcut CoT:
> | num perturbed exemplars |    0   |    1   |   2   |    4   |    8    |
> |:----------------------------------|:------:|:------:|:-----:|:------:|:------:|
> | ChatGPT                           | 0.79 | 0.80 | 0.79 | 0.79 | 0.80 |
> | LLaMA-2 13B                    | 0.30 | 0.30 | 0.28 | 0.31 | 0.31 |
> | LLaMA-2 7B                      | 0.20 | 0.20 | 0.20 | 0.21 | 0.22 |
>
> GSM8K Typo LTM:
> | num perturbed exemplars |    0   |    1   |   2   |    4   |    8    |
> |:----------------------------------|:------:|:------:|:-----:|:------:|:------:|
> | ChatGPT                           | 0.74 | 0.73 | 0.73 | 0.72 | 0.74 |
> | LLaMA-2 13B                    | 0.26 | 0.26 | 0.25 | 0.28 | 0.26 |
> | LLaMA-2 7B                      | 0.16 | 0.16 | 0.17 | 0.17 | 0.15 |
>
> GSM8K Synonym LTM:
> | num perturbed exemplars |    0   |    1   |   2   |    4   |    8    |
> |:----------------------------------|:------:|:------:|:-----:|:------:|:------:|
> | ChatGPT                           | 0.70 | 0.69 | 0.70 | 0.70 | 0.71 |
> | LLaMA-2 13B                    | 0.24 | 0.24 | 0.26 | 0.26 | 0.27 |
> | LLaMA-2 7B                      | 0.15 | 0.15 | 0.14 | 0.18 | 0.17 |
>
> GSM8K Repetition LTM:
> | num perturbed exemplars |    0   |    1   |   2   |    4   |    8    |
> |:----------------------------------|:------:|:------:|:-----:|:------:|:------:|
> | ChatGPT                           | 0.75 | 0.75 | 0.76 | 0.76 | 0.77 |
> | LLaMA-2 13B                    | 0.29 | 0.29 | 0.28 | 0.28 | 0.30 |
> | LLaMA-2 7B                      | 0.18 | 0.18 | 0.19 | 0.20 | 0.20 |
>
> GSM8K Shortcut LTM:
> | num perturbed exemplars |    0   |    1   |   2   |    4   |    8    |
> |:----------------------------------|:------:|:------:|:-----:|:------:|:------:|
> | ChatGPT                           | 0.59 | 0.67 | 0.70 | 0.74 | 0.70 |
> | LLaMA-2 13B                    | 0.28 | 0.28 | 0.26 | 0.26 | 0.24 |
> | LLaMA-2 7B                      | 0.16 | 0.16 | 0.18 | 0.20 | 0.23 |
>
> Strategy QA Typo CoT:
> | num perturbed exemplars |    0   |    1   |   2   |    4   |    8    |
> |:----------------------------------|:------:|:------:|:-----:|:------:|:------:|
> | ChatGPT                           | 0.64 | 0.64 | 0.66 | 0.66 | 0.67 |
> | LLaMA-2 13B                    | 0.54 | 0.54 | 0.57 | 0.58 | 0.52 |
> | LLaMA-2 7B                      | 0.53 | 0.53 | 0.55 | 0.49 | 0.52 |
>
> Strategy QA Synonym CoT:
> | num perturbed exemplars |    0   |    1   |   2   |    4   |    8    |
> |:----------------------------------|:------:|:------:|:-----:|:------:|:------:|
> | ChatGPT                           | 0.63 | 0.61 | 0.62 | 0.62 | 0.64 |
> | LLaMA-2 13B                    | 0.52 | 0.51 | 0.51 | 0.53 | 0.53 |
> | LLaMA-2 7B                      | 0.53 | 0.48 | 0.53 | 0.54 | 0.56 |
>
> Strategy QA Repetition CoT:
> | num perturbed exemplars |    0   |    1   |   2   |    4   |    8    |
> |:----------------------------------|:------:|:------:|:-----:|:------:|:------:|
> | ChatGPT                           | 0.67 | 0.67 | 0.67 | 0.68 | 0.67 |
> | LLaMA-2 13B                    | 0.57 | 0.57 | 0.57 | 0.57 | 0.57 |
> | LLaMA-2 7B                      | 0.54 | 0.54 | 0.54 | 0.54 | 0.54 |
>
> Strategy QA Shortcut CoT:
> | num perturbed exemplars |    0   |    1   |   2   |    4   |    8    |
> |:----------------------------------|:------:|:------:|:-----:|:------:|:------:|
> | ChatGPT                           | 0.76 | 0.76 | 0.76 | 0.77 | 0.76 |
> | LLaMA-2 13B                    | 0.62 | 0.61 | 0.63 | 0.61 | 0.61 |
> | LLaMA-2 7B                      | 0.61 | 0.61 | 0.60 | 0.58 | 0.57 |
>
> Consistent with the previous results on ChatGPT, LLaMA 7B generally seems to favor having more perturbed examples than fewer. Also consistent with earlier results, CoT is more robust against perturbations than LTM prompting. We propose to replace Figs 3 and 4 with the data from the new experiments shown above (rendered as bar plots) and add additional text to Sec 5 to better contextualize the results, mentioning that the models are more sensitive to perturbations on GSM8K than StrategyQA. We will also qualify the claims in the Introduction and Discussion sections accordingly.
>
> [1] Mor Geva, Daniel Khashabi, Elad Segal, Tushar Khot, Dan Roth, & Jonathan Berant (2021). Did Aristotle Use a Laptop? A Question Answering Benchmark with Implicit Reasoning Strategies. TACL.
>
> ## Answer to the Reviewer's Question
>
> Q: The author claims that one of the goals is to explore empirical approaches that may improve their robustness, but I did not see much discussion on this. Could you provide more information on it?
>
> A: We are happy to clarify. The goal of the experiment where we vary the number of perturbed examples is to test whether including perturbations in the in-context prompt can make the model more robust against those perturbations. Indeed we find that for some perturbations (notably Shortcut with LTM prompting), perturbed in-context examples can significantly improve the robustness of the model. We propose to include this in Sec 6.

---

### Meta-Review · Area_Chair_oz2A · 2023-09-18

**Recommendation:** 4

**Metareview:**

To assess the robustness of Large Language Models on multi-hop arithmetic reasoning tasks, the manuscript proposes a systematic approach that encompasses both lexical and semantic perturbations. All reviewers agree that the comparative analysis of distinct prompting methods, coupled with various proposed perturbations, offers significant insights. To further improve the quality of current draft, reviewers suggest 1) expanding to other reasoning task for wider generalization, 2) investigating different perturbations such as styles and orders, and 3) incorporating human evaluations as a supplement to the automated assessments. As some of these points are addressed during rebuttal, it is encouraged to properly integrate them into the revised manuscript.

---

### Decision · Program_Chairs · 2023-10-07

**Decision:**

Accept-Main

**Comment:**

To assess the robustness of Large Language Models on multi-hop arithmetic reasoning tasks, the manuscript proposes a systematic approach that encompasses both lexical and semantic perturbations. All reviewers agree that the comparative analysis of distinct prompting methods, coupled with various proposed perturbations, offers significant insights. To further improve the quality of current draft, reviewers suggest 1) expanding to other reasoning task for wider generalization, 2) investigating different perturbations such as styles and orders, and 3) incorporating human evaluations as a supplement to the automated assessments. As some of these points are addressed during rebuttal, it is encouraged to properly integrate them into the revised manuscript.